# Comparative Proteomic Analyses of Poorly Motile Swamp Buffalo Spermatozoa Reveal Low Energy Metabolism and Deficiencies in Motility-Related Proteins

**DOI:** 10.3390/ani12131706

**Published:** 2022-07-01

**Authors:** Runfeng Liu, Xingchen Huang, Qinqiang Sun, Zhen Hou, Weihan Yang, Junjun Zhang, Pengfei Zhang, Liangfeng Huang, Yangqing Lu, Qiang Fu

**Affiliations:** State Key Laboratory for Conservation and Utilization of Subtropical Agro-Bioresources, Guangxi University, Nanning 530004, China; liurunfeng101@163.com (R.L.); hxcyyds6688@126.com (X.H.); dwfzs101@163.com (Q.S.); dwfzs1122@163.com (Z.H.); yangweihan2021@163.com (W.Y.); zjjyyds6688@163.com (J.Z.); zpfyyds101@163.com (P.Z.); hlfyyds101@126.com (L.H.)

**Keywords:** buffalo, sperm motility, proteomics, LC-MS/MS, tandem mass targeting

## Abstract

**Simple Summary:**

Proteomic analysis of normal and poorly motile buffalo sperm indicates that the formation of poorly motile buffalo sperm is associated with inefficient energy metabolism, decreased sperm protamine proteins, a lack of motility-related proteins, and changes in tail structural proteins. The results provide clues for finding molecular mechanisms for poor motility of buffalo sperm.

**Abstract:**

The acquisition of mammalian sperm motility is a main indicator of epididymal sperm maturation and helps ensure fertilization. Poor sperm motility will prevent sperm cells from reaching the fertilization site, resulting in fertilization failure. To investigate the proteomic profiling of normal and poorly motile buffalo spermatozoa, a strategy applying liquid chromatography tandem mass spectrometry combined with tandem mass targeting was used. As a result, 145 differentially expressed proteins (DEPs) were identified in poorly motile spermatozoa (fold change > 1.5), including 52 upregulated and 93 downregulated proteins. The upregulated DEPs were mainly involved in morphogenesis and regulation of cell differentiation. The downregulated DEPs were involved with transport, oxidation-reduction, sperm motility, regulation of cAMP metabolism and regulation of DNA methylation. The mRNA and protein levels of PRM1 and AKAP3 were lower in poorly motile spermatozoa, while the expressions of SDC2, TEKT3 and IDH1 were not correlated with motility, indicating that their protein changes were affected by transcription or translation. Such changes in the expression of these proteins suggest that the formation of poorly motile buffalo spermatozoa reflects a low efficiency of energy metabolism, decreases in sperm protamine proteins, deficiencies in motility-related proteins, and variations in tail structural proteins. Such proteins could be biomarkers of poorly motile spermatozoa. These results illustrate some of the molecular mechanisms associated with poorly motile spermatozoa and provide clues for finding molecular markers of these pathways.

## 1. Introduction

Mammalian sperm cells, the male gametes, are produced in the testes through a complex process of spermatogenesis that transforms diploid spermatogonia into motile haploid spermatozoa. Following ejaculation, sperm cells travel a long distance in the female reproductive tract and swim against a series of obstacles such as low vaginal pH, the cervix and the presence of macrophages in the uterus. These obstacles serve as selection mechanisms to prevent abnormal spermatozoa from reaching the oocyte. During their transit through the reproductive tract, spermatozoa undergo a complex process for hyperactivation, referred to as capacitation [1,2]. In humans, infertility is a multifactorial pathology and more than 50% of cases are attributed to the male partner [3]. In particular, sperm motility is a prerequisite for successful fertilization.

Sperm movement requires energy that is supported by two metabolic pathways involved in ATP synthesis: oxidative phosphorylation and glycolysis [4]. Furthermore, there are many proteins that affect sperm motility. Thus, spermatozoa from mice lacking the protein transaldolase are infertile because they lack progressive motility [5]. Junctional adhesion molecule A (JAM-A) is essential for normal sperm motility in the oviduct. Studies on JAM-A gene-deficient mice have shown that both progressive and hyperactivated sperm motility were significantly impaired [6]. Various studies have focused on finding biomarkers that affect sperm motility using proteomics approaches [3,7,8,9,10]. Some important proteins are differentially expressed between human spermatozoa with normal motility and those from men with idiopathic asthenozoospermia [11]. Thus, protein tyrosine phosphatase non-receptor type 14 was lower in both mRNA and protein levels from moderately motile than from progressively motile spermatozoa [12]. In humans, seminal plasma and spermatozoa undergo protein exchange following ejaculation, which regulate the material transport process needed for successful fertilization [13].The seminal plasma proteins such as ZAG and AAT have also been implicated in regulating sperm motility [14,15]. There have been studies on human spermatozoa and seminal plasma for the identification of biomarkers of asthenozoospermia [11,16,17]. Thus, experiments have shown that ENOL protein expression was correlated with sperm motility, and this might be a potential marker for low sperm motility [16]. In buffalo, some studies have discovered important biomarkers affecting sperm development through proteomic techniques [17]. Huang et al. found that ADRM1, K21 ubiquitination may be very important for buffalo spermatogenesis [18]. In a subsequent study, buffalo testis development was found to be regulated by CABYR phosphorylation through phosphoproteome analysis [19]. Zhang et al. found that the ubiquitination sites of PSMA3 and RAF1 may play an important role in the development of buffalo epididymal sperm [20].

Swamp buffalo (*Bubalus bubalis*) play a significant role in the agricultural economy of many developing countries. The species is adapted to hot-humid tropical climatic conditions and comparatively resistant to bovine diseases. Despite these merits, buffalo have relatively poor reproductive efficiency irrespective of their location throughout the world. Buffalo exhibit many of the known reproductive disorders including sperm abnormal motility. Sperm are one of the transcriptional silencing cells; the physiological regulatory and motility acquisition rely on protein expression. To our knowledge, there are limited studies on buffalo sperm proteomics. To investigate the differentially expressed proteins (DEPs) in normal and poorly motile spermatozoa, an integrated proteomic approach—liquid chromatography tandem mass spectrometry (LC–MS/MS) combined with tandem mass tagging (TMT)—was applied to profile the sperm proteome. The goal of this research was to reveal any significant changes in proteins and identify the molecular pathway differences between poorly motile and normally motile forms. This study will provide clues for finding candidate biomarkers associated with buffalo sperm motility.

## 2. Materials and Methods

All buffalos were maintained according to the principles of the Guangxi guide for the care and experimental use of laboratory animals.

### 2.1. Sperm Preparation

The buffalo semen samples were collected from the Livestock Breed Improvement Station (Guangxi Province, China). Normal semen (≥80% motile forms) and poorly motile semen (≤40%) samples were collected (*n* = 12 individuals in each group, age between 3 and 10 years) via an artificial vagina. Two groups of fresh semen samples were pooled and washed three times with phosphate-buffered saline. To remove cell debris and abnormal spermatozoa, a density gradient Percoll (Sigma, St. Louis, MO, USA, Cat No. P4937) solution was prepared in a 5 mL Eppendorf tube (2.5 mL of 90% Percoll solution under 2.5 mL of 45% Percoll solution). The sperm pellets were layered on the top of the Percoll gradient and then were centrifuged at 400× *g* for 30 min at 4 °C. Subsequently, the pelleted sperm were resuspended in sperm diluent and sperm motility assessment analysis was performed immediately. Each 20 μL of sperm was injected into preheated 20 μL chambers and analyzed by Computer Assisted Sperm Analyzer (CASA) (IVOS Ⅱ, Hamilton Thorne, Beverly, MA, USA). Kinematic parameters were evaluated with the Hamilton Thorne CEROS CASA software (version 14.0, Beverly, MA, USA). A total of 30 video frames was acquired at 60 Hz, and 13 tracking points were set in each chamber [21].

### 2.2. Protein Extraction

Protein extraction from normal and poorly motile buffalo spermatozoa was carried out as described [22]. Briefly, cells were ruptured by ultrasonication for 30 min after being resuspended in lysis buffer (7M urea, 2M thiourea, 4% CHAPS, 1% DTT and 1% *v*/*v* protease inhibitor). Supernatants were collected by centrifuging at 10,800× *g* for 30 min. Subsequently, proteins were precipitated with precooled acetone (−20 °C) overnight. Finally, the protein concentration was determined using Bradford assays.

### 2.3. Protein Digestion and TMT Labeling

Aliquots of 100 µg of proteins from normal and poorly motile sperm samples were resuspended in 45 µL of 100 mM triethylammonium bicarbonate solution plus 0.5% SDS. Proteins were reduced by adding 5 µL of 200 mM Tris (2-carboxyethyl) phosphine for 1 h at 55 °C, followed by adding 5 µL of 375 mM iodoacetamide for 30 min in darkness at room temperature for alkylation. The protein samples were digested enzymatically by adding 2.5 µg trypsin at 37 °C for 16 h. Finally, the peptides from normal and poorly motile spermatozoa were labeled with TMT-129 or TMT-126, respectively, according to the manufacturer’s instructions (TMT 6-plex Label Reagents kit; Thermo Fisher Scientific, Rockford, IL, USA, Cat. No. 90064).

### 2.4. Peptide Prefractionation and On-Line LC–MS/MS Analysis

The peptide mixture was redissolved in buffer A (98% Milli-Q H_2_O, 2% acetonitrile, ACN, pH 10.0) and then fractionated using normal pH reversed-phase liquid chromatography with an HPLC system (Waters, Allience e6295, Milford, MA, USA) connected to an XBridge C18 column (4.6 mm × 250 mm, 3.5 µm, 130 Å) at a flow rate of 0.5 mL/min. Elution was done using 100% buffer A for 10 min. The linear gradient started from 5% to 35% buffer B (98% ACN, 2% ddH_2_O, pH 10.0) for 55 min, 35% to 100% buffer B for 5 min, and was finally rinsed with 100% buffer B for 5 min. The fractions were collected and pooled into eight aliquots, then desalted using Zip-Tip C18 Tips (Millipore, St. Louis, MO, USA; Cat. No. 87782). All peptides were analyzed using an LTQ-Orbitrap Elite hybrid mass spectrometer (Thermo Fisher Scientific, Bremen, Germany) interfaced with an Easy-nLC 1000 nano liquid chromatography system (Thermo Fisher Scientific, Odense, Denmark). MS analysis was carried out in data-dependent acquisition (DDA) mode in the scan range of *m*/*z* 350–1800, and survey scans were acquired in the Orbitrap analyzer at a mass resolution of 60,000 at 400 *m*/*z*. Ten of the most intense precursor ions were selected for MS^2^ in the normal energy collision dissociation (HCD) mode in the linear ion trap. The dynamic exclusion parameters were as follows: exclusion count, 2; exclusion time, 40 s. Siloxane ions were used for internal calibration (*m*/*z*, 445.1200).

### 2.5. Data Analysis

The data results identified by mass spectrometry were searched against the bovine database (Download from NCBInr, Taxonomy ID: 9913), and a threshold of False Distribution Rate (FDR) was set at 0.05. The biological functions of DEPs were determined according to information available at the UniProt Knowledgebase website (http://www.uniprot.org, accessed on 6 March 2021). Gene Ontology (GO) and Kyoto Encyclopedia of Genes and Genomes (KEGG; https://www.genome.jp/kegg/, accessed on 9 March 2021) analyses were carried out using the KOBAS 2.0 server (http://kobas.cbi.pku.edu.cn, accessed on 10 March 2021) as described in [23]. Protein interaction networks were searched using String 10.0 (https://string-db.org, accessed on 14 March 2021) and visualized using Cytoscape 3.2 software (Accessed on 15 March 2021). Statistical analysis was performed using GraphPad Prism 8.

### 2.6. Western Blot Analysis

DEPs from normal and poorly motile buffalo sperm samples were validated by Western blot analysis. Aliquots of 20 µg of each protein sample were loaded to 12% SDS–PAGE, and then the proteins in the gel were transferred to PVDF membranes (Bio-Rad, USA, Cat No. 1620177). PVDF membranes were blocked in TBST buffer (10 mM Tris, 150 mM NaCl, 0.05% Tween 20, 5% skim milk, pH 7.6) for 2 h, incubated with primary antibodies (Abcam, Cambridge, UK, Cat. No. 108051, Cat No.66978, Cat No.205884, Cat No.170856; Beyotime, Beijing, China, Cat No. AF7950, Cat No.AF0210) in 5% skim milk overnight at 4 °C, then washed using TBST three times (10 min each). Subsequently, the PVDF membranes were incubated with horseradish peroxidase-conjugated second antibodies (ZSGB-BIO, Beijing, China; Cat. No. zb-2308, 1:500 dilution) at room temperature for 1 h. Signals detected by a molecular imager (Bio-Rad, USA, PharosFX) were analyzed using BioRad Quantity One software to obtain the gray value of the proteins. GAPDH was used as loading control.

### 2.7. Quantitative Reverse Transcription Polymerase Chain Reaction (RT–qPCR)

Total RNAs were extracted from normal and poorly motile sperm samples using RNeasy Midi kits (Qiagen, Hilden, Germany) and reverse-transcribed to cDNA using PrimeScript RT Master Mix (TaKaRa, Dalian, China). Amplification was performed using a LightCycler 480 real-time PCR instrument (Roche, Basel, Switzerland) using SYBR Green Supermix (Bio-Rad, Hercules, CA, USA). Forward and reverse primers, template and ddH_2_O were added up to a total volume of 20 µL. The procedure was as follows: the first DNA sequence was obtained by reverse transcription at 50 °C for 30 min, then 30 cycles of 95 °C for 2 min, 94 °C for 20 s, 60 °C for 20 s, and 70 °C for 20 s. The *GAPDH* gene was used as an internal control. The relative expression levels from three experiments were analyzed using 2^−ΔΔCT^ values. The primer sequences for RT–qPCR are listed in Appendix A.

## 3. Results

### 3.1. Proteomic Profile Analysis

Normal and poorly motile buffalo sperm samples were separated as described (Section 2.1). CASA analysis showed that the normal samples had 79.6 ± 7.1% motile cells versus 17.2 ± 6.7% in the poorly motile samples (Means ± SD, *p* < 0.01; Figure 1). In proteome profile analysis, proteins derived from normal and poorly motile sperm were labeled using a TMT-based strategy and analyzed by LC–MS/MS. As shown in Figure 2, 1148 and 1171 proteins were labeled and identified in the two replicates, respectively. Venn diagram analysis indicates that 981 of these proteins overlapped in the two groups of sperm samples. The proteomic data are listed in Appendix A. Among all of the labeled proteins, 145 were identified as DEPs between the normal and poorly motile sperm samples (fold change ≥ 1.5). Compared with the normal sperm samples, poorly motile sperm samples included 52 upregulated and 93 downregulated proteins (Appendix A).

### 3.2. Bioinformatics Analysis

During the bioinformatics analysis, a *FDR* < 0.05 was used as the confidence interval. Subcellular classification analysis of 145 DEPs showed that they were mainly distributed in the cytoplasm (25%), extracellular exosomes (19%), mitochondria (12%), extracellular regions (10%), cytoskeleton (7%), mitochondrial inner membrane (4%) and flagellum (3%) (Figure 3). In biological processes, nearly half of the proteins had no functional GO annotations. Among the annotated DEPs, nine categories were correlated directly with sperm development, the most frequent being transport (19), oxidation–reduction processes (9), flagellum morphogenesis (4) and microtubule-based processes (9). Among the 19 proteins related to sperm transport, 14 proteins (74%) were downregulated in poorly motile sperm samples, including lactotransferrin (LTF), lipoprotein carrier (LCN5, LCN12), phosphatidyl alcohol transporter (PITPNA) and dynein (DYN). The reduced level of dynein indicated that the low transport efficiency of metabolites was one of the reasons for inadequate sperm motility. Furthermore, there were abnormal expressions of two proteins associated with DNA methylation (KDM1B and RLF), indicating that the DNA methylation levels of poorly motile and normal spermatozoa were inconsistent. Testicle-specific histone 2B (TH2B) and protamine 1 (PRM1) were downregulated in poorly motile sperm samples (Figure 4a). In terms of molecular function, DEPs were involved in 10 categories, including protein homodimerization activity (9), structural molecule activity (7), protein domain-specific binding (5), protein *N* terminus binding (4), lyase activity (4), ATPase activity (4), enzyme inhibitor activity (3), carboxylic ester hydrolase activity (3), chaperone binding (3) and nitric oxide synthase regulator activity (2). Calmodulin-1 (CALM1) not only participates in the processes of spermatogenesis and capacitation by regulating Ca^2+^ concentration, but also regulates the activity of nitric oxide. The downregulated expression of Calm1 in poorly motile sperm samples might be associated with the insufficient regulation of active nitrogen (Figure 4b).

### 3.3. KEGG Pathway Analysis of DEPs

To check the potential pathways of these DEPs, KEGG pathways were analyzed; 145 DEPs were mapped successfully to 66 KEGG orthology (Ko) IDs using UniProt (http://www.uniprot.org/uploadlists/, accessed on 6 March 2021), and 122 relative pathways were obtained (Appendix A). Some important pathways are shown in Table 1. The most involved was the metabolic pathway (ko01100). The identification of thermogenesis (ko04714) and oxidative phosphorylation (ko00190) pathways confirms that the maintenance of sperm motility requires a large amount of energy.

### 3.4. MGI Reproductive Phenotype Analysis

The mouse is a model for the study of mammalian gene function, and there is a complete Mouse Genome Informatics (MGI) website (http://www.informatics.jax.org, accessed on 14 March 2021). This was used to explore the correlations between the buffalo DEPs we identified with known reproductive disorders [24]. The results show that the DEPs were associated with sixty-seven reproductive system disorders, and eight of these genes were involved with male sterility (MP: 0001925). Some important phenotypic terms are listed in Table 2. Thus, protein phosphatase 1 (Ppp1cc) was related to 25 phenotypes such as a small testis (MP: 0001147), small seminiferous tubules (MP: 0001153), abnormal spermiogenesis (MP: 0001932) and abnormal sperm physiology (MP: 0004543), suggesting that a Ppp1cc-mediated signaling pathway plays a biological role in various stages of sperm development.

### 3.5. Protein–Protein Interaction (PPI) Analysis

To obtain the PPI network of the DEPs, STRING software was used to perform an online analysis. Cytoscape software was used to visualize images and the plug-ins Centiscape 2.2 and ClusterONE were used for PPI network analysis. As shown in Figure 5, these DEPs are mainly involved in physiological and biochemical sperm processes such as flagellum motion, energy metabolism and tail structure of sperm.

### 3.6. Western Blot and RT–qPCR Analysis

To validate the results of quantitative proteomics, the expression levels of six im-portance DEPs were analyzed by the Western blot densitometric readings (Appendix A). As shown in Figure 6, where GAPDH was used as loading control, the protein levels had no significant variations in different lanes. Among them, the expression levels of PRM1, AKAP3, CCDC40, TEKT3 and IDH1 were higher in normal than in poorly motile sperm samples. The expression level of SDC2 was higher in the poorly motile than in the normal samples. The results of Western blot analysis are consistent with quantitative proteomic analysis.

The RT–qPCR results show that the *Prm1*, *Sdc2*, *Akap3*, *Tekt3* and *Idh1* genes could be amplified and quantified successfully in spermatozoa, but this was not the case for *Ccdc40*. The mRNA levels of *Prm1* and *Akap3* were lower in poorly motile than in normal sperm samples, indicating that these transcriptional expressions were consistent with protein levels. However, there were no significant differences in the expression levels of *Sdc2*, *Tekt3* and *Idh1* genes between normal and poorly motile sperm samples, indicating that these were not correlated with the expression of mRNA, which might have arisen from post-transcriptional or translation processes.

## 4. Discussion

Proteomic approaches are used to explore the composition and change of all proteins in a specific cell or tissue at a holistic level, which is an important part of genomic activity [23,25]. Multidimensional separation of samples, mass spectrometry identification and bioinformatics analysis are the three technical pillars of proteomics research [26]. The rapid development of proteomics has relied on large-scale throughput separation and analysis techniques. There are two main methods for proteome separation: 2-DE and LC–MS. In a previous study, 2-DE combined with MALDI-TOF/TOF mass spectrometry was used by us for evaluating the proteomics of normal and poorly motile sperm samples. Three proteins, ODF2, ATP5A1 and SUCLG2, were identified and validated. These results help in understanding the molecular mechanisms associated with poorly motile sperm samples [9]. However, the LC–MS approach has almost completely replaced 2-DE technologies and has become the “gold standard” for proteomics research [27]. Quantitative proteomics strategies based on this approach allow the monitoring of large-scale dynamic changes in protein expression levels and identifying functional proteins [28].

Here, a quantitative proteomic analysis of TMT reagents combined with LC–MS/MS was performed to investigate the differential protein expression profiles of normal and poorly motile sperm samples. Sperm function is important in buffalo reproduction, and therefore it is important to study its characteristics and direct research toward the identification of new potential biomolecular markers for fertility and infertility. In this context, determining what distinguishes motile from immotile spermatozoa at the proteome level is important to determine which proteins and cellular pathways play a role in sperm motility. As a result, 1441 proteins were identified, including 145 DEPs (change fold > 1.5), of which 52 were upregulated and 93 were downregulated in poorly motile sperm samples. From the overall trend of protein expression changes, the DEPs in poorly motile sperm sample were generally downregulated, which is consistent with a previous report [29]. Functional analysis (pathway enrichment and PPI network analysis) of proteomic data provides a comprehensive profile of the molecular and cellular pathways affected in buffalo sperm motility. The DEPs were related to: (1) oxidative stress, (2) energy metabolism, (3) cytoskeleton integrity and (4) canonical signaling pathways associated with spermatogenesis (Figure 7).

### 4.1. Abnormal Nuclear Chromatin of Poorly Motile Buffalo Spermatozoa

Chromosomal DNA is compacted into the cell nucleus primarily by histones in somatic cells and by protamines in sperm nuclei. Sperm DNA integrity is regulated by the expression and abundance of sperm proteins. During spermiogenesis, nuclear remodeling and condensation are associated with the sequential displacement of histones by transition proteins and then by PRM1 and PRM2 [30]. The relationships between the mouse *Prm1* and *Prm2* genes is not well understood, and gene KO experiments have shown that both *Prm1* and *Prm2* are required for normal spermiogenesis and male fertility in mice [31]. In addition, the correct proportion of the two protamines in mice is critical for maintaining the integrity of sperm chromatin [32]. Alterations in the composition and structural organization of sperm chromatin (involving PRM1 and PRM2) can affect both fertilization and early embryogenesis [33]. Here we found that the level of PRM1 in poorly motile sperm samples was downregulated, and the expression levels of four histones, H1C, H2A, H2B and H3H, were all downregulated, which could cause loosening of the chromatin structure and affect the genetic stability of the genome in poorly motile spermatozoa. IZUMO1 protein is a chromatin remodeling regulator, which was had higher expression in poorly motile sperm forms, impairing chromatin condensation and inducing head abnormality as well as poor sperm motility. Additionally, as one of the molecular chaperones, heat shock proteins (HSPs) are in involved in stage-specific and developmentally regulated spermiogenesis. Bioinformatics analysis revealed that HSPs were predominately associated with posttranslational modification, protein folding and ubiquitination degradation, and lipid and nucleic acid metabolism. A 26S protease was found to be upregulated, suggesting that proper regulation of protein folding and ubiquitination degradation mechanisms are essential for sperm maturation and motility acquisition.

### 4.2. Decreased Mitochondrial Energy Metabolism in Poorly Motile Sperm Samples

The mitochondria serves as the power source of sperm, and are hence essential for sperm function. A previous study [34] showed that the production of ATP in motile spermatozoa mainly depended on glycolysis and mitochondrial respiration, and that an abnormal expression of associated enzymes could affect sperm motility. Comparing the DEPs from normal and poorly motile sperm samples, we found that five enzymes in the oxidative phosphorylation pathway were significantly downregulated, indicating that poor sperm motility was associated with insufficient mitochondrial energy production. Among these enzymes, the cytochrome oxidases COX6B and COX6C are located in the nuclei of mammalian cells and on the inner membrane of mitochondria, and are transported to mitochondria by ribosomes [35]. Then, they are assembled together with other subunits to form complexes, COXs, which are the only mitochondrial enzymes that can transfer electrons to oxygen molecules [36,37].We speculate that the decreased expression of COX6B/COX6C resulted in inadequate COX enzyme activity, which suppressed the production of mitochondrial ATP and consequently led to poor buffalo sperm motility. Seminal oxidative stress is a major cause of male infertility. An excessive number of reactive oxygen species (ROS) results in lipid peroxidation of the sperm membrane and damages the sperm DNA integrity, thereby compromising normal sperm functions. Under oxidative stress conditions, proteins associated with carbohydrate metabolic pathways, such as gluconeogenesis and glycolysis, and protein modification are compromised in poorly motile sperm samples. Furthermore, stress response as well as cellular, metabolic and regulatory pathways are dysregulated with increased levels of ROS.

### 4.3. Relationship between Tail Structure Proteins and Motion in Poorly Motile Spermatozoa

Spermiogenesis is the last step of spermatogenesis, including nuclear condensation in the sperm head and formation of the flagellum [38]. Progressive motility of spermatozoa entering the female reproductive tract mainly relies on flagellum activity for successful sperm transport and fertilization of oocytes [39]. An intact flagellum is essential for the spermatozoon to reach the fertilization site (the oviduct) against a series of obstacles. Therefore, we can explore the causes of poor motility through changes of proteins in the sperm tail. Tektins (TEKTs) are intermediate filament-like proteins localized in cilia, flagella, basal bodies and centrioles [40]. They are highly coiled-coil molecules [41]. Mammals possess five TEKT proteins (TEKT1 to 5), and each isoform can be found in spermatozoa [40,41,42,43]. They are mainly involved in the formation of the inner axoneme, and maintain the stability of the central doublet microtubules [44]. *Tekt3*-null mice produce spermatozoa with reduced progressive motility and increased structural bending defects of the flagellum, suggesting that *Tekt3* might be involved in idiopathic asthenozoospermia in humans [45]. Subsequently, TEKT3 was found to be present at the equatorial segment of the acrosome in sperm heads. Therefore, TEKT3 might not only work as a flagellum constituent required for stability, but might also be involved in acrosomal function, such as the acrosome reaction or sperm–oocyte fusion [46]. In our study, the expression of TEKT3 tended to be downregulated in poorly motile sperm samples, in line with previous studies.

The axonemal microtubules are surrounded by proteins such as dynein arms and dynein regulatory complexes. Dynein arms are the molecular motors of microtubule doublet sliding, acting to complete the flagellum beating cycle by repeated binding and release [47]. CCDC40 (coiled-coil domain containing protein) is expressed in tissues that contain motile cilia, and mutation of *Ccdc40* results in cilia with reduced ranges of motility. Importantly, CCDC40 deficiency causes primary ciliary dyskinesia (PCD) [48,49]. Spermatozoa from humans with mutations of the *Ccdc40* gene show defects of central microtubules and medial dynein arms [50]. Sui et al. identified some mutant alleles in the *Ccdc40* gene, which altered the protein sequence and resulted in ultrastructural defects in the microtubule structure of cilia. These defects can lead to human male infertility [51]. In our work, the expression levels of DYNLL1 and DYNLT1 were downregulated and the expression levels of DCNT2 and CCDC40 were upregulated in poorly motile buffalo sperm samples, which is consistent with the hypothesis that dynein arm disorders can lead to decreased sperm motility. Therefore, the *Tekt3*, *Dynll1*, *Dynlt2*, *Dcnt2* and *Ccdc40* identified in our study could be used as candidate genes for PCD.

At present, there has been a tremendous increase in research on sperm proteomics, which has facilitated the advancement of sperm physiology research. In the next studies, the research focus should be geared towards identification and clinical validation of a panel of proteins that could serve as biomarkers for specific male infertility. Extensive phosphorylation proteomic research would shed light on the potential mechanism of sperm maturation. Moreover, correlation analysis of the proteome and transcriptome also provides a more reliable basis for evaluating, the exact mechanism that regulates sperm motility.

## 5. Conclusions

In general, our study confirmed the relationship between proteomic changes and the maintenance of buffalo sperm motility. Thus, 145 DEPs affecting sperm motility were identified. Poor sperm motility was associated with weak energy metabolism, decreases in sperm protamine proteins, deficiencies in motility-related proteins, and variations in flagellum structural proteins. These potential candidate biomarkers provide clues for finding molecular mechanisms associated with poor mammalian sperm motility.

## Figures and Tables

**Figure 1 animals-12-01706-f001:**
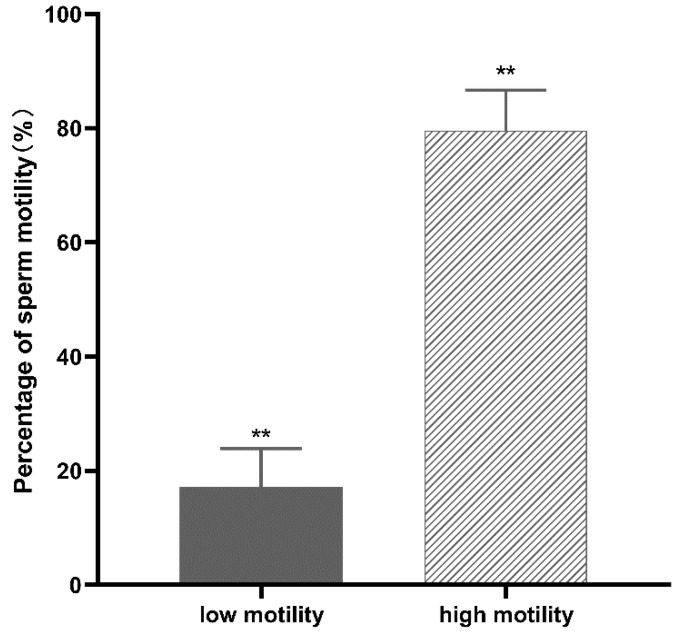
Comparison of the percentages of motile spermatozoa in normal and poorly motile sperm samples. ** *p* < 0.01.

**Figure 2 animals-12-01706-f002:**
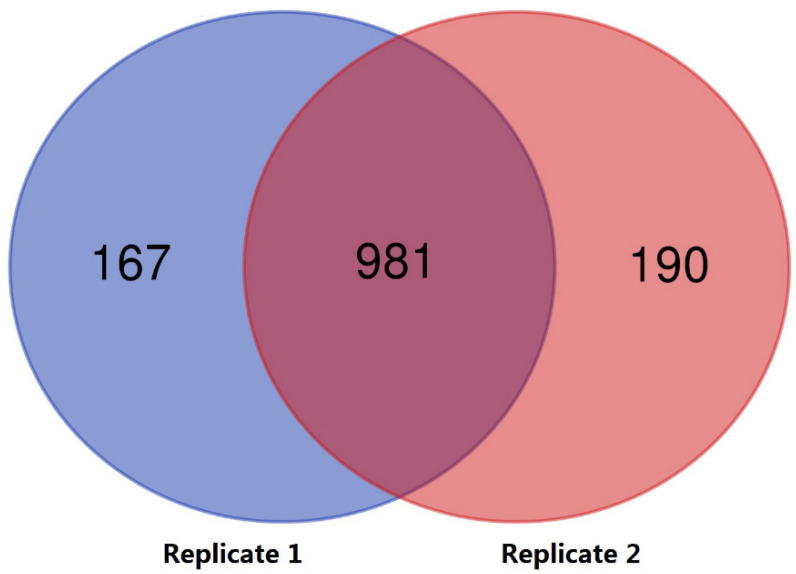
Venn diagram of sperm proteomic replicates. Replicate 1 detected 1148 proteins (blue), Replicates 2 detected 1171 proteins (red), and 981 proteins (purple) were replicated twice.

**Figure 3 animals-12-01706-f003:**
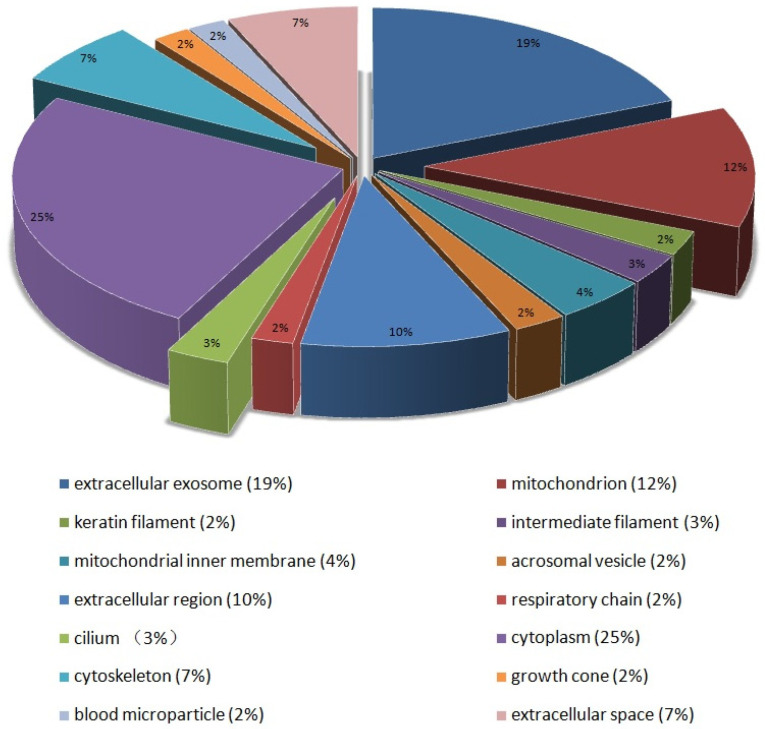
Subcellular classification of DEPs.

**Figure 4 animals-12-01706-f004:**
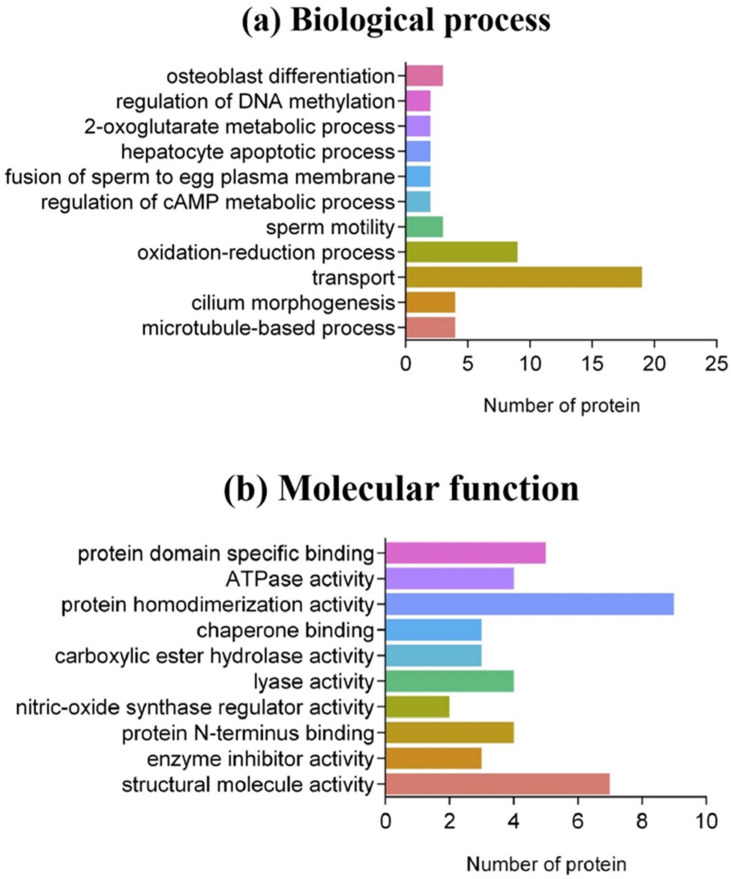
GO analysis of DEPs in poorly motile sperm samples. (**a**) Bar graph showing the numbers of proteins involved in biological processes; (**b**) bar diagram showing the numbers of proteins involved in molecular functions.

**Figure 5 animals-12-01706-f005:**
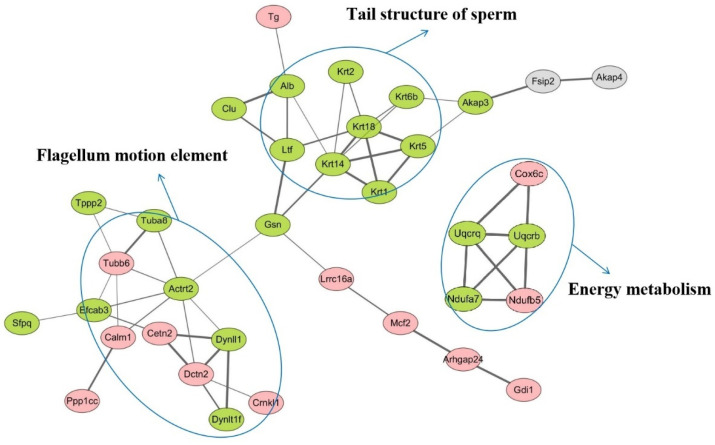
The protein–protein interaction network analysis of DEPs. Note: Only proteins with known interactions were listed. The red and green nodes represent upregulated and downregulated proteins, respectively.

**Figure 6 animals-12-01706-f006:**
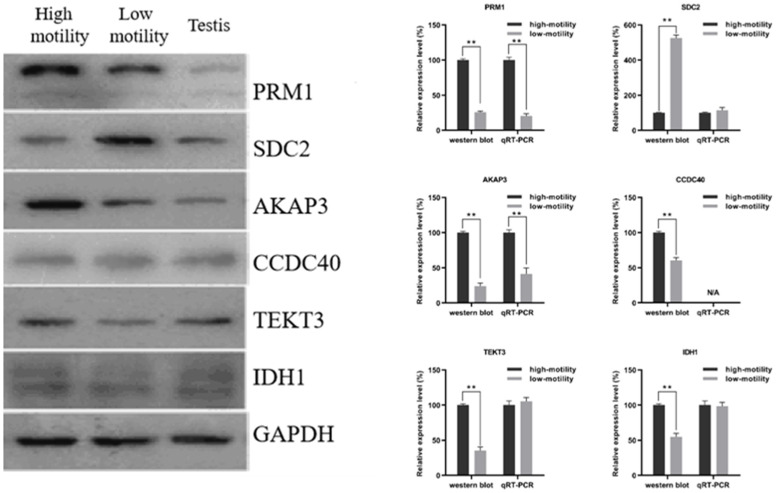
Western blotting and RT–qPCR validation of DEP expression profiles, ** *p* < 0.01.

**Figure 7 animals-12-01706-f007:**
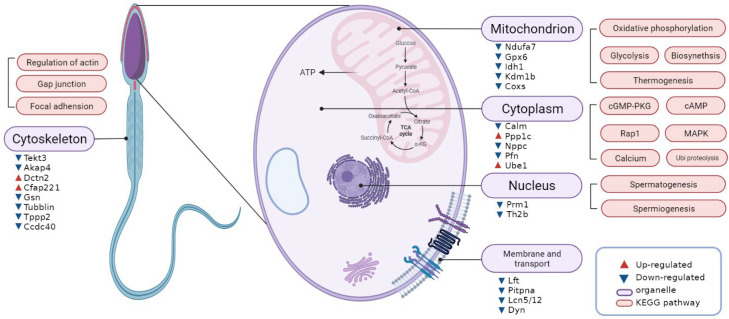
The subcellular location and regulatory pathways of DEPs.

**Table 1 animals-12-01706-t001:** KEGG pathway analysis of DEPs.

KEGG ID	Pathway Name	Protein Counts	Protein Names
ko01100	Metabolic pathways	11	IDH1; QCR7; QCR8; ALD0; E4.1.3.4; ATPeF0F6; COX6C; NDUFA7; NDUFB5; ECHS1; ASRGL1
ko04714	Thermogenesis	8	QCR7; QCR8; ATPeF0F6; COX6C; NDUFA7; NDUFB5; NDUFAF5; COA1
ko00190	Oxidative phosphorylation	6	QCR7; QCR8; ATPeF0F6; COX6C; NDUFA7; NDUFB5;
ko01110	Biosynthesis of secondary metabolites	4	IDH1; ALDO; ECHS1; ASRGL1
ko04022	cGMP-PKG signaling pathway	3	CALM; PPP1C; NPPC
ko04979	Cholesterol metabolism	2	VAPB; NPC2
ko04218	Cellular senescence	2	CALM; PPP1C
ko04120	Ubiquitin mediated proteolysis	2	UBE1; ELOC
ko04024	cAMP signaling pathway	2	CALM; PPP1C
ko01230	Biosynthesis of amino acids	2	IDH1; ALDO
ko04015	Rap1 signaling pathway	2	CALM; PFN
ko04810	Regulation of actin cytoskeleton	2	PFN; PPP1C
ko04540	Gap junction	2	TUBA; TUBB
ko04510	Focal adhesion	1	PPP1C
ko04013	MAPK signaling pathway	1	PFN
ko01212	Fatty acid metabolism	1	ECHS1
ko00010	Glycolysis/Gluconeogenesis	1	ALDO
ko04020	Calcium signaling pathway	1	CALM
ko00072	Synthesis and degradation of ketone bodies	1	E4.1.3.4

**Table 2 animals-12-01706-t002:** Mammalian phenotype terms for DEPs in buffalo spermatozoa.

Mammalian Phenotype ID	Mammalian Phenotype Term	Gene Names
MP: 0001925	male infertility	*Akap4*, *Ppp1cc*, *Ddx25*, *Izumo1*, *Adam1a*, *S1c9c1*, *Odf1*
MP: 0002675	asthenozoospermia	*Ppp1cc*, *Tekt3*, *Adam1a*, *S1c9c1*, *Odf1*
MP: 0003984	Embryonic growth retardation	*Lig3*, *C1qbp*, *Rbpj*
MP: 0005389	Abnormal Reproductive system phenotype	*Rdh11*, *Akap4*, *Ppp1cc*, *Cd46*, *Tekt3*, *Mcf2*, *Nppc*, *Odf1*
MP: 0005410	Abnormal fertilization	*Cd46*, *Adam1a*
MP: 0009238	Coiled sperm flagellum	*Akap4*, *Odf1*, *Dynll1*
MP: 0009832	Abnormal sperm mitochondrial sheath morphology	*Akap4*, *Ppp1cc*, *Odf1*
MP: 0009836	Abnormal sperm principal piece morphology	*Akap4*, *Ppp1cc*
MP: 0011092	Embryonic lethality	*Zfp335*, *Slc3a2*, *Rbpj*, *C1qbp*, *Glrx3*, *Gsn*, *Lig3*, *Hmgc1*

## Data Availability

The data presented in this study are available on request from the corresponding author. The data are not publicly available due to privacy restrictions.

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
