# Peer review of "Comparative Proteomic Analyses of Poorly Motile Swamp Buffalo Spermatozoa Reveal Low Energy Metabolism and Deficiencies in Motility-Related Proteins"

_animals, 2022, doi:10.3390/ani12131706_

Round 1
Reviewer 1 Report
The conclusions are too obvious

Author Response
Dear reviewer:
Thank you for your review of our manuscript. We appreciated the concerns and useful comments and suggestions from you. Now we have revised our manuscript according to your opinions. At this time, we have re-submitted the revised manuscript through the MDPI website, and we look forward to the feedback from you again.
We have accepted your suggestion. We have made changes in the foreword to explain the environment in which the buffalo lives and the reasons for their low fertility, and to further explain the purpose of our research. In Materials and methods, relevant parameters for CASA analysis are supplemented. In the discussion, we have made significant revisions and added the discussion content based on signaling pathways to enrich the content. In the updated Supplementary file, we have supplemented with relevant information on the WB images. We checked the writing of the article as well as formatting issues and made a comprehensive revision.
As to details, Please find the revised manuscript for your approval. All revise were highlighted and marked using the function "Track changes". Alternatively, please check the point-by-point responses to your review opinion as follow.
The detailed point-to point response was described as followed:
Point 1: The conclusions are too obvious
Response 1:Thank you very much for your suggestion. The purpose of this study was to reveal any significant changes in proteins between poor and normal forms of exercise. This research will provide clues for finding biomarkers associated with buffalo sperm motility. Some proteins found in this study have been reported in other species, but need to be further explored in buffalo. We added new content to the Discussion to enrich the findings.
Swamp buffalo (Bubalus bubalis) has a significant role in the agricultural economy of many developing countries. The species are adapted to hot-humid tropical climatic condi-tions and more resistant to bovine diseases. Despite these merits, but buffalo have rela-tively poor reproductive efficiency irrespective of their location throughout the world. Buf-falo exhibits many of the known reproductive disorders including sperm abnormal motil-ity. Sperm are one of transcriptional silencing cells, the physiological regulatory and mo-tility acquisition rely on protein expression. To our knowledge, there are limited studies on buffalo sperm proteomics. To investigate the differentially expressed proteins (DEPs) in normal and poorly motile spermatozoa, an integrated proteomic approach, liquid chro-matography tandem mass spectrometry (LC–MS/MS) combined with tandem mass tag-ging (TMT) was applied to profile the sperm proteome. The goal of this research was to reveal any significant changes in proteins and identify the molecular pathways between poorly motile and normally motile forms. This study will provide clues for finding can-didate biomarkers associated with buffalo sperm motility.
Thanks again for your suggestion.

Reviewer 2 Report
Question 1.-
Did the semen analayze after or before centrifugation?
Line 85- The buffalo semen samples were collected from the Livestock Breed Improvement 85 Station (Guangxi Province, China). Normal semen (80% motile forms) and poorly mo-86 tile semen (40%) samples were collected
Line 93- Subsequently, the sperm collected from 93 the monolayer were analyzed using a computer assisted sperm analysis (CASA) system 94 (Hamilton Thorne, USA, IVOS â…¡TM) to measure sperm motility.
Coments
Some errors are found, there are with yelow marks

Author Response
Dear reviewer:
Thank you for your review of our manuscript. We appreciated the concerns and useful comments and suggestions from you. Now we have revised our manuscript according to your opinions. At this time, we have re-submitted the revised manuscript through the MDPI website, and we look forward to the feedback from you again.
All your suggestion were accepted. We have made major revision in the Introduction section to explain the environment in which the buffalo lives and the possible reasons for their low reproductive efficiency, and to further explain the purpose of our research. In Materials and methods section, relevant parameters for CASA analysis were supplemented. In the discussion, we have made significant revisions and new figure to complete the discussion content based on signaling pathways. In the updated Supplementary file, we have supplemented with relevant information on the WB images. Some minor errors were corrected carefully.
As to details, Please find the revised manuscript for your approval. All revise were highlighted and marked using the function "Track changes". Alternatively, please check the point-by-point responses to your review opinion as follow.
The detailed point-to point response was described as followed:
Point 1: Did the semen analayze after or before centrifugation?
Line 85- The buffalo semen samples were collected from the Livestock Breed Improvement 85 Station (Guangxi Province, China). Normal semen (80% motile forms) and poorly mo-86 tile semen (40%) samples were collected
Line 93- Subsequently, the sperm collected from 93 the monolayer were analyzed using a computer assisted sperm analysis (CASA) system 94 (Hamilton Thorne, USA, IVOS â…¡TM) to measure sperm motility.
Response 1:To remove some impurities and the influence of necrotic sperm, we performed CASA analysis after centrifugation. And supplemented the relevant parameters of CASA analysis.
The buffalo semen samples were collected from the Livestock Breed Improvement Station (Guangxi Province, China). Normal semen (80% motile forms) and poorly motile semen (40%) samples were collected (n = 12 individuals in each group, age between 3 and 10 years) via an artificial vagina. Two groups of fresh semen samples were pooled and washed three times with phosphate-buffered saline. To remove cell debris and abnormal spermatozoa, a density gradient Percoll (Sigma, USA, Cat No.P4937) solution was prepared in a 5 mL Eppendorf tube (2.5 mL of 90% Percoll solution under 2.5 mL of 45% Percoll solution). The sperm pellets were layered on the top of the Percoll gradient and then were centrifuged at 400 g for 30 min at 4°C. Subsequently, the pelleted sperm were resuspended in sperm diluent and immediately performed for sperm motility as-sessment analysis. Each 20 μL of sperm were injected into preheated 20 μL chambers and analyzed by Computer Assisted Sperm Analyzer (CASA) (IVOSâ…¡, Hamilton Thorne, USA). Kinematic parameters were evaluated with the Hamilton Thorne CEROS CASA software (version 14.0). 30 video frames were acquired at 60 Hz, and 13 tracking points were set in each chamber.

Reviewer 3 Report
Comments to authors
The current study by Liu et al. on “Comparative proteomic analyses of poorly motile swamp buffalo spermatozoa reveal low energy metabolism and deficiencies in motility-related proteins” is an interesting study. Even there are no scientific novel points in this research, it can be accepted from a technical point of view. However, it is recommended to submit to a technology targeting journal rather than animals. And the technical comments are provided below:
- The introduction is unclearly descibed and the purpose of the study is not clear. At end of the introduction, the authors explained that Swamp buffalo (Bubalus bubalis) is adapted to hot-humid tropical climatic conditions but has poor reproductive efficiency. Are your study is focused on the alteration of the motility-related proteins? It is too confusing.
- The information on Primary antibodies was not found in the draft manuscript.
- The protein profiling method is quite rough. Therefore, normally, researchers select up a minimum of>2-fold chanced proteins. I think > 1.5-fold is not enough differential expressed level without individual confirmation of each profiled protein by Western blot.
- The western blot image was too poor. In addition, uncropped images were not distinguished which is the representative image for whom. The western blot data was not trustable images to readers.
- The discussion type is not normal. If the authors want to determine and discuss the categories of the data. You can merge the result and discussion part.
- Almost discussion was performed based on references, not physiological data. Perhaps, since the authors performed mobility analysis through CASA, it is necessary to provide various mobility-related parameters, and discussions based on this are necessary.
- Identified proteins were already identified proteins related to sperm motility. Therefore, the present study was not novel. However, the authors provided several bioinformatics data. Thus, the discussion is needed based on the signaling pathway, it is more important in the present study.
- It is necessary to revise that emphasizes the unity of the overall thesis form.
Author Response
Dear reviewer:
Thank you for your review of our manuscript. We appreciated the concerns and useful comments and suggestions from you. Now we have revised our manuscript according to your opinions. At this time, we have re-submitted the revised manuscript through the MDPI website, and we look forward to the feedback from you again.
All your suggestion were accepted. We have made major revision in the Introduction section to explain the environment in which the buffalo lives and the possible reasons for their low reproductive efficiency, and to further explain the purpose of our research. In Materials and methods section, relevant parameters for CASA analysis were supplemented. In the discussion, we have made significant revisions and new figure to complete the discussion content based on signaling pathways. In the updated Supplementary file, we have supplemented with relevant information on the WB images. Some minor errors were corrected carefully.
As to details, Please find the revised manuscript for your approval. All revise were highlighted and marked using the function "Track changes". Alternatively, please check the point-by-point responses to your review opinion as follow.
The detailed point-to point response was described as followed:
The current study by Liu et al. on “Comparative proteomic analyses of poorly motile swamp buffalo spermatozoa reveal low energy metabolism and deficiencies in motility-related proteins” is an interesting study. Even there are no scientific novel points in this research, it can be accepted from a technical point of view. However, it is recommended to submit to a technology targeting journal rather than animals. And the technical comments are provided below:
Point 1: The introduction is unclearly descibed and the purpose of the study is not clear. At end of the introduction, the authors explained that Swamp buffalo (Bubalus bubalis) is adapted to hot-humid tropical climatic conditions but has poor reproductive efficiency. Are your study is focused on the alteration of the motility-related proteins? It is too confusing.
Response 1:We have accepted your suggestion. We have made changes in the introduction to explain the conditions in which the buffalo lives and the reasons for their low fertility. Buffaloes exhibit many known reproductive disorders, including abnormal sperm motility. Sperm is one of the transcriptionally silent cells, whose physiological regulation and motility acquisition depend on protein expression.
Our research focuses onmotility-related proteins. The reproductive efficiency of buffalo is poor and we speculate that it may be related to the expression of some proteins in sperm. In our study, we found some differentially expressed proteins, and from the general trend of protein expression changes, DEP was generally down-regulated in spermatozoa with poor motility. Decreased mitochondrial energy metabolism in sperm samples with poor motility may lead to reduced motility performance. The relationship between sperm tail structural proteins and motility of poor motility, the expression of some tail structural proteins decreased in sperm with poor motility, resulting in incomplete sperm tail development, resulting in a decrease in motility performance.
Point 2: The information on Primary antibodies was not found in the draft manuscript.
Response 2: Thank you for your suggestion. We have added product information about primary antibodies in the material and methods. The antibodies were obtained from two sources as follow, (Abcam, Cambridge, UK, Cat. No. 108051 , 66978, 205884, 170856; Beyotime, Beijing, China, Cat No. AF7950, AF0210)
Point 3: The protein profiling method is quite rough. Therefore, normally, researchers select up a minimum of>2-fold chanced proteins. I think > 1.5-fold is not enough differential expressed level without individual confirmation of each profiled protein by Western blot.
Response 3: Thank you for your suggestion. We refer to some literatures on the choice of the difference fold, and find that >1.5 folder is a more convincing choice, and even >1.3 in some articles. Second, the TMT or iTRAQ quantification approach based on the peak area of precursor and fragment ion acquired from mass spectrometry. This strategy have some characteristic that minimized the change folder of proteins. In order to acquire complete information of DEPs in sperm motilitya more appropriate fold changes > 1.5 or < 0.67 were set as the threshold. In most published articles, the fold change value is taken into consideration.
Please check the following articles:
Lin L, Wang Y, Srinivasan R, Zhang L, Song H, Song Q, Wang G, Lin X. Quantitative Proteomics Reveals That the Protein Components of Outer Membrane Vesicles (OMVs) in Aeromonas hydrophila Play Protective Roles in Antibiotic Resistance. J Proteome Res. 2022 Jun 8. doi: 10.1021/acs.jproteome.2c00114. Epub ahead of print. PMID: 35674493.
Xu J, Guan X, Jia X, Li H, Chen R, Lu Y. In-depth Profiling and Quantification of the Lysine Acetylome in Hepatocellular Carcinoma with a Trapped Ion Mobility Mass Spectrometer. Mol Cell Proteomics. 2022 Jun 7:100255. doi: 10.1016/j.mcpro.2022.100255. Epub ahead of print. PMID: 35688384.
Point 4: The western blot image was too poor. In addition, uncropped images were not distinguished which is the representative image for whom. The western blot data was not trustable images to readers.
Response 4: Thank you for your suggestion. We reprocessed the images and annotated detailed information in the uncropped images. The raw images were uploaded as a supplementary file with optical density value.
The poor image maybe attribute to the species reactivity of primary antibody, since the antibody was purified for mouse, human or other species. The specificity of antibody is not good enough. However, the commercial corporation have not product the specific antibody used for buffalo. If necessary, we will validate the protein expression using parallel reaction monitoring (PRM) method (PRM validation could product the ratio of protein expression without antibody)
Point 5: The discussion type is not normal. If the authors want to determine and discuss the categories of the data. You can merge the result and discussion part.
Almost discussion was performed based on references, not physiological data. Perhaps, since the authors performed mobility analysis through CASA, it is necessary to provide various mobility-related parameters, and discussions based on this are necessary.
Identified proteins were already identified proteins related to sperm motility. Therefore, the present study was not novel. However, the authors provided several bioinformatics data. Thus, the discussion is needed based on the signaling pathway, it is more important in the present study.
Response 5: Thank you for your suggestion. We have determined serval parameters of the CASA analysis such as sperm concentration, motility, progressive forward et al. In this manuscript, the motility is the mainly issue. other data in the CASA analysis are not shown in the result section. We've added something completely new to the discussion.
Semen |
Sperm concentration(108/ml) |
Sperm motility(%) |
|
Sperm abnormality(%) |
High motility |
8.06±3.33a |
80.63±7.06a |
|
12.37±3.08a |
Low motility |
7.98±2.67 |
37.35±6.56a |
|
14.68±6.26a |
Note: Between the high motility(n=12) and low motility(n=12), different letters in the same column mean a significant difference (p<0.01 ) and the same letters in the same column means no significant difference(p>0.05)
We have discussed based on signaling pathways to enrich the content, and in the Figure 7, we can clearly see the signaling pathways of DEPs. The proteins located in cytoskeleton associated with regulation of actin skeleton, gap junction and focal adhension, mitochondrion serve as an energy power source, which relate to oxidative phosphorylation, glycolysis and other energy metabolism. Many DEPs located in cytoplasm participated in many signaling pathways, such as cGMP-PKC, Rap1, MAPK and Calcium ion regulation pathway. Please check
the detailed content in Line 341-349, Line 366-373 and Line 410-416.
In general, this study is the primary research on sperm proteomics, in future study, we should focus on identification and clinical validation of a panel of proteins that could serve as biomarkers for specific male infertility. Extensive phosphorylation proteomic is necessary for sperm maturation.

Reviewer 4 Report
Although not original, the study produces amd makes available to the reader some new information on proteomic approach to semen testing. The study in its present form is therefore worthy of publication. Authors are encouraged though to check for errors in the written English. See for example the very beginning of manuscript (see Simple Summary, second line)
Author Response
Dear reviewer:
Thank you for your review of our manuscript. We appreciated the concerns and useful comments and suggestions from you. Now we have revised our manuscript according to your opinions. At this time, we have re-submitted the revised manuscript through the MDPI website, and we look forward to the feedback from you again.
All your suggestion were accepted. We have made major revision in the Introduction section to explain the environment in which the buffalo lives and the possible reasons for their low reproductive efficiency, and to further explain the purpose of our research. In Materials and methods section, relevant parameters for CASA analysis were supplemented. In the discussion, we have made significant revisions and new figure to complete the discussion content based on signaling pathways. In the updated Supplementary file, we have supplemented with relevant information on the WB images. Some minor errors were corrected carefully.
As to details, Please find the revised manuscript for your approval. All revise were highlighted and marked using the function "Track changes". Alternatively, please check the point-by-point responses to your review opinion as follow.
The detailed point-to point response was described as followed:
Point 1: Although not original, the study produces amd makes available to the reader some new information on proteomic approach to semen testing. The study in its present form is therefore worthy of publication. Authors are encouraged though to check for errors in the written English. See for example the very beginning of manuscript (see Simple Summary, second line)
Response 1: Thank you for your suggestion. We have checked the English writing in the article and made changes.

Round 2
Reviewer 3 Report
Although the authors tried to respond to arisen queries from me, the major issues were not resolved yet.
Author Response
Dear reviewer:
